# Preparation, Antimicrobial Properties under Different Light Sources, Mechanisms and Applications of TiO_2_: A Review

**DOI:** 10.3390/ma15175820

**Published:** 2022-08-24

**Authors:** Changyu Shang, Junyu Bu, Cui Song

**Affiliations:** Marine College, Shandong University, Weihai 264209, China

**Keywords:** photocatalytic antimicrobial, TiO_2_, photocatalyst, mechanisms

## Abstract

Traditional antimicrobial methods, such as antibiotics and disinfectants, may cause adverse effects, such as bacterial resistance and allergic reactions. Photocatalysts based on titanium dioxide (TiO_2_) have shown great potential in the field of antimicrobials because of their high efficiency, lack of pollution, and lack of side effects. This paper focuses on the antimicrobial activity of TiO_2_ under different light sources. To improve the photocatalytic efficiency of TiO_2_, we can reduce electron-hole recombination and extend the photocatalytic activity to the visible light region by doping with different ions or compounds and compounding with polymers. We can also improve the surface properties of materials, increase the contact area with microorganisms, and further enhance the resistance to microorganisms. In addition, we also reviewed their main synthesis methods, related mechanisms, and main application fields to provide new ideas for the enhancement of photocatalytic microorganism performance and application popularization in the future.

## 1. Introduction

Microbial pollution has become an important factor that threatens human health and can produce various toxins [1]. Pathogenic microorganisms can cause human diseases [2,3], and some microorganisms can cause food spoilage [4], molds in clothes [5], degraded water quality [6], and even biological corrosion of buildings [7], causing certain losses to human society.

To reduce the adverse effects of microorganisms, disinfectants, antibiotics, and other methods are used to kill microorganisms, but there are also many problems in using these methods. Disinfectants can produce disinfection byproducts (DBPs) with high toxicity [8,9], leading to asthma, allergic rhinitis, and other respiratory symptoms [10] and corrosion effects [11]. The excessive use of antibiotics will produce new drug-resistant strains. Drug-resistant strains will periodically explode, increasing the incidence and becoming a major challenge in the global public health field [12,13].

Among these antimicrobial strategies, photocatalytic antimicrobials can realize the efficient utilization of light energy with fewer side effects which has received a wide range of attention. The photocatalyst plays an important role in the photocatalytic antimicrobial process. Because of this reason, research on photocatalysts and photocatalytic antimicrobials has been increasing overall (Figure 1). Among many photocatalytic materials, TiO_2_ and its composites are the most widely studied photocatalysts due to their low cost, stability, and easy preparation. The photoelectrochemical properties of TiO_2_ have a history of nearly 100 years, and many breakthrough discoveries have also promoted its development. In 1938, Goodeve et al. [14] studied the photochemical bleaching of ‘Chlorazol Sky Blue’ with TiO_2_. In 1972, Fujishima and Honda discovered for the first time that TiO_2_ could be used as an electrode in the photocatalytic splitting of water [15]. Without any applied voltage, water could be decomposed into oxygen and hydrogen by visible radiation, thus realizing the transformation from optical energy to chemical energy. In 1997, Wang et al. [16] prepared TiO_2_-coated glass. After UV irradiation, the structure of the TiO_2_ coating showed highly amphiphilic characteristics, which made the TiO_2_-coated glass antifogged and self-cleaning. 

In the field of photocatalytic antimicrobials, TiO_2_ and its composites have excellent killing effects on *Escherichia coli, Staphylococcus aureus, Candida albicans, Aspergillus niger,* and other microorganisms They can be widely used in sewage treatment, food packaging, self-cleaning fabrics, antimicrobial coatings, and other fields [17,18,19]. In this paper, the main preparation methods of TiO_2_, antimicrobial activity and improvement strategies under ultraviolet (UV) light and visible light, antimicrobial mechanisms, and applications are reviewed to provide ideas and directions for future research.

## 2. TiO_2_ and Its Composites Materials

### 2.1. Synthesis Methods

At present, there are many synthesis methods for TiO_2_ photocatalysts. The synthesis method and conditions have a great influence on the performance of the photocatalyst. Fully understanding the characteristics of photocatalyst synthesis methods is beneficial to improve not only the production efficiency but also the application.

#### 2.1.1. Sol–Gel Method

The sol–gel method is one of the most commonly used methods for the synthesis of TiO_2_. This method does not require a high reaction temperature. It can control the texture and surface properties of the material by changing the pH value, temperature, and reaction time [20]. It has unique advantages in the composition and structure control of inorganic materials and hybrid materials [21,22], is the simplest, has a low cost, and is suitable for laboratory research [23]. The sol–gel method uses alcohol or water as the medium to prepare the final product through hydrolysis, polycondensation, aging, drying, and thermal decomposition [20,24] (Figure 2A). Vargas et al. [25] obtained ceramic powder by the sol–gel method and then controlled the subsequent heat treatment temperature to ensure the purity of the oxide and the nanometric size of the particles. The synthesized powders were amorphous up to a temperature T < 350 °C, with a particle size of 100 nm. Their SEM and TEM images are shown in Figure 2B. Tahmasebizad et al. [26] also used the sol–gel method to synthesize Cu and N-codoped TiO_2_ photocatalytic coating materials, which can be activated by visible light.

#### 2.1.2. Microwave-Assisted Method

The microwave-assisted method is commonly used with other methods as a way to enhance photocatalytic performance. The microwave-assisted method can improve the dispersion of different phases, accelerate the reaction rate, and modify the components in the sample [27,28,29,30]. For example, Ates et al. [30] synthesized a target sample within 10 min under microwave irradiation (Figure 2C). Lu et al. [31,32] prepared two different metal-doped catalysts, TiO_2_-Ag (*p* = 600 W, T = 140 °C) and TiO_2_-Sm (*p* = 450 W, T = 170 °C), by a microwave hydrothermal method. The SEM images of the two catalysts are shown in Figure 2D. Experiments show that microwave irradiation can improve the photocatalytic activity of TiO_2_ by uniformly heating the reaction system from inside to outside and increasing the number of ·OH on the surface of TiO_2_-Ag/Sm.

#### 2.1.3. Green Synthesis Method

In recent years, many studies have been conducted on green synthesis methods, which are based on natural raw materials and are environmentally friendly [33,34]. Green synthesis methods are often used in conjunction with other methods. The synthesis process shown in Figure 2E is a combination of the green synthesis method and coprecipitation method [35]. Sagadevan et al. [36] synthesized TiO_2_ nanoparticles using Myristica fragrans seed extract. The synthesized TiO_2_ can be activated by UV light and has antibacterial activity against *Klebsiella pneumoniae* and *S. aureus*.

#### 2.1.4. Hydrothermal Method

The hydrothermal method is used to synthesize products with relatively small crystal structures, which is usually carried out in autoclaves with water as a medium at high temperature and high pressure [37]. Based on one-dimensional TiO_2_ obtained by calcining Ti foil at 800 °C, Sun et al. [38] prepared nanotube films by the hydrothermal method. The growth process of the nanotube structure from rutile TiO_2_ on Ti foil is investigated by the morphology change of TiO_2_ films prepared with different hydrothermal durations. TiO_2_ with different structures was prepared by a hydrothermal method using butyl titanate and absolute ethanol as raw materials [39]. Researchers found that anatase TiO_2_ is formed when the hydrothermal time is 12 h, TiO_2_ forms mixed crystals of anatase and rutile when the hydrothermal time is 24 h, and rutile TiO_2_ is formed when the hydrothermal time is 36 h. Their SEM images are shown in Figure 2F.

#### 2.1.5. Atomic Layer Deposition and Magnetron Sputtering Methods

Atomic layer deposition (ALD) and magnetron sputtering (MS) methods are often used to prepare photocatalyst composite films. They can realize the doping [40] and deposition [41,42] of metal on photocatalysts or the loading of photocatalysts on other substrates [43,44], which can improve the optical properties of materials. The MS method is based on the principle of physical deposition, and the surface of the prepared film is flatter [45]. This method is also often influenced by the shape of the substrate, and film is deposited only on surfaces parallel to the target surface [46]. The composition and structure of the ALD method are highly controllable, and the thickness of the prepared film is uniform, which is not affected by the shape of the material. The reaction is generally divided into four steps: (1) Precursor supply, (2) Purge, (3) Reactant supply, (4) Purge (Figure 2G) [47]. The required thickness is obtained through multiple cycles. Abdulagatov et al. [44] made a V-doped TiO_2_ film on polypropylene (PP) hernia meshes by using the thermal ALD method after more than 100 cycles. The film thickness is 38 nm, having good photocatalytic antibacterial performance (Figure 2H).

**Figure 2 materials-15-05820-f002:**
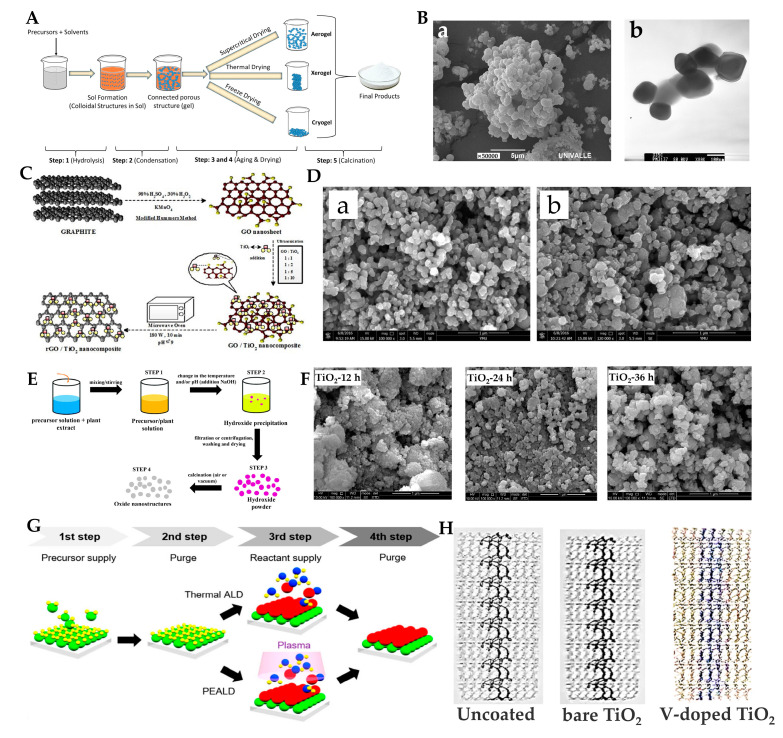
(**A**) Steps involved in the sol–gel process to synthesize MONPs [20]; (**B**) scanning electron microscopy (SEM) (**a**) and transmission- electron microscopy (TEM) (**b**) images of antimicrobials synthesized by the sol–gel method and thermally treated at 350 °C for 2 h [25]; (**C**) illustration of rGO/TiO_2_ nanocomposite formation [30]; (**D**) SEM images of TiO_2_-Sm (**a**) and TiO_2_-Ag (**b**) [31,32]; (**E**) schematic diagram showing all steps involved in a generic green synthesis mediated by plant extract using the coprecipitation method [35]; (**F**) SEM images of different TiO_2_ samples [39]; (**G**) illustration of ALD [47]; (**H**) different PP mesh samples [44].

### 2.2. Antimicrobial Activity under UV Light

UV irradiation is a common source of energy for TiO_2_ photocatalysts. TiO_2_ photocatalysts have a wide band gap energy of 3.2 eV and can be activated under UV light [17]. Table 1 lists some TiO_2_ antimicrobial materials that can be activated under UV light.

Heloisa N et al. [48] deposited three kinds of crystalline TiO_2_ films on commercially pure titanium (cp-Ti) by MS. As shown in Figure 3A, four groups of TiO_2_ films were obtained: (1) machined cp-Ti (control); (2) A-TiO_2_ (anatase); (3) M-TiO_2_ (mixture of anatase and rutile); and (4) R-TiO_2_ (rutile). Researchers collected volunteers’ oral saliva as a source of multispecies biofilms and observed the inhibition by different films (Figure 3B). The photocatalytic performance of the three kinds of crystalline TiO_2_ films increases with increasing UV irradiation time. Figure 3C shows the colony forming units (log_10_ CFU/cm^2^) of the total number on the surfaces after 1 h of UVA light exposure according to groups. The results showed that both A-TiO_2_ and M-TiO_2_ showed significant antibacterial activity after 1 h of UV irradiation (*p* < 0.001), and the number of bacteria decreased by approximately 99% and 99.9%, respectively, while the R-TiO_2_ film had no antibacterial effect on the multispecies biofilms (*p* < 0.05).

The photocatalytic properties of many materials are improved after modification. For example, Moongraksathum et al. [49] used the sol–gel method to prepare a series of TiO_2_ nanocomposite films with different silver contents (Ag/TiO_2_). Under UVA illumination for 1 h, the antibacterial effect of the film on *E. coli* can be higher than 99.99%, while that of TiO_2_ could only reach 16.33%. The killing effect on influenza A virus (H1N1) and enterovirus also reached nearly 100%. Silica is a kind of amorphous material with stable chemical properties. Chen et al. [50] synthesized a silica carrier by the gel method and then synthesized a composite material (TiO_2_@SiO_2_) by the hydrolysis method. SiO_2_ can increase the specific surface area and the ability of TiO_2_ to adsorb bacteria. The surface of normal *E. coli* was smooth, but the surface of TiO_2_@SiO_2_ treated *E. coli* became rough and even showed small holes, indicating that the cell membrane of *E. coli* was damaged (Figure 4A). As shown in Figure 4B,C, with increasing TiO_2_ component content, TiO_2_@SiO_2_ has excellent antibacterial properties under UVA light. Pessoa et al. [43] deposited TiO_2_ films on polyurethane (PU) and polydimethylsiloxane (PDMS) by the ADL method and studied the photocatalytic antibacterial effect on *Candida albicans*. Even in the absence of UV irradiation, TiO_2_/PU could inhibit the conversion of *C. albicans* to mycelium. Under UV irradiation, the CFU of TiO_2_/PDMS samples can be reduced to 80%, which is much higher than 25% in the control group.

Hayashi et al. [51] hydrothermally synthesized TiO_2_ nanosheets (NS) with highly oriented structures. By changing the ratio of F/Ti in the reaction raw material, a series of samples with different morphologies were obtained: For *Streptococcus mutans*, the sample with the best performance can make the bacterial survival rate less than 10% under UV irradiation, while that of the nonirradiated group was more than 40%. It maintained a bacterial survival rate below 40%, even at a low concentration (0.1 mg/L).

**Table 1 materials-15-05820-t001:** Antimicrobial effects of different TiO_2_ materials under UV light.

Material	Grain Size: nm	Microorganism	Light Source	Antimicrobial Activity	Reference
A-TiO_2_, M-TiO_2_, R-TiO_2_	Approximately 300 nm	*Streptococcus sanguinis, Actinomyces naeslundii, and Fusobacterium nucleatum*	UVA 2 × 15 W (λ = 350 nm and intensity = 1.62 mW/cm^2^)	Approximately 99.9% for M-TiO_2_ and 99% for A-TiO_2_, R-TiO_2_ had no antimicrobial activity	[48]
Ag-TiO_2_	1–3 (Ag)	*E. coli*, the influenza A virus (H1N1), and enterovirus	UVA (20 W/cm^2^)	>99.9% (1 wt% Ag/TiO_2_)	[49]
TiO_2_(NS1.0, NS1.2, NS1.5, NS1.8, and NS2.0 with different F/Ti ratios)	200–600 (length), 6–20 (thickness)	*S. mutans*	365 nm light (2.5 mW/cm^2^)	Approximately 90% (NS1.0)	[51]
Cu/TiO_2_ nonwoven fabric (NWF)	20 (TiO_2_), 1–5 (Cu)	HuNoV genogroup II genotype 4 (HuNoV GII.4)	373-nm UVA-LED source	The optimum treatment conditions for inactivating the HuNoV GII.4 droplets were as follows: Cu:TiO_2_ ratio of 1:7.7 and the use of a 373 nm UVA-LED source for 48.08 min	[52]
Polyurethane-acrylate-Ag/TiO_2_	TiO_2_: 30(length), 3–4(diameter), Ag: 6(diameter)	*E. coli*	36 W mercury lamp (wavelength >245 nm)	Approximately 97%	[53]
Fe-TiO_2_	12–15	*E. coli*	Black light blue lamp with the major fraction of irradiation occurring at 365 nm	Almost 100% (0.5% Fe-doped, 10 min)	[54]
SiO_2_@ TiO_2_	20.49	*E. coli*	UVA (30 W)	99.3%	[50]

### 2.3. Antimicrobial Activity under Visible Light

UV light only accounts for 5% of sunlight, far less than 45% for visible light, while pure TiO_2_ can only be activated under a UV light source because of its shortfall response to visible light and the expeditious recombination rate of the photogenerated electron–hole pairs which greatly limits its utilization of solar energy [55]. If its absorption spectrum can move to the visible light region, full use of the readily available energy of sunlight can be made, and the photocatalytic efficiency can be improved [56]. Table 2 lists some different TiO_2_ antimicrobial materials that can be activated under visible light. Many methods have been adopted to improve the utilization rate of light energy and strengthen the antimicrobial effect of materials [57].

#### 2.3.1. TiO_2_-Metal Composites

TiO_2_Ag [58], Pt [59], Cu [58,60], Fe [61], Cr [62], V [44], and other metals can improve the photocatalytic antimicrobial activity of TiO_2_ by forming composites through modification, doping, and surface deposition.

Gomez-Polo et al. [62] found that the antibacterial ability of Cr-doped TiO_2_ under visible light can reach more than 200 times that under dark conditions. Cr promotes the shift of the absorption spectrum from UV to visible light, and the distribution is uniform in TiO_2_, which increases the efficiency of receiving optical energy. Au and Pt NPs can be used as intermediate carriers for the valence electron transition to the conduction band of TiO_2_, extending the photocatalytic activity to the visible region [59,63,64], which improves the photocatalytic antibacterial efficiency of materials. The V-doped TiO_2_ photocatalytic coating prepared by the ACL method can effectively inhibit the adhesion of *S. aureus* and *E. coli*. Doping V makes TiO_2_ produce a more intense redox reaction under visible light, killing microorganisms more effectively and preventing the formation of biofilms [44]. Figure 5A shows the antibacterial properties of TiO_2_ doped with different metals against *E. coli* in the dark (a) and under light (b). The antibacterial properties of some materials, such as Ag-TiO_2_ and Cu-TiO_2_, have better performance under visible light than in the dark [58]. Moon et al. [59] deposited Au and Pt coatings on TiO_2_ nanotubes (Figure 5B). These coatings have different bacterial adhesion abilities, and the colony-forming unit (CFU) values of the control group were lower than those of the light-free condition under visible light irradiation, especially under visible light irradiation at 470 nm wavelength. Figure 5C shows the results of the agar diffusion test of *S. aureus* (a) and CFUs of *S. aureus* under visible light irradiation at 470 nm and 600 nm (b).

#### 2.3.2. Nonmetal Doping

In addition to the metals mentioned above, TiO_2_ is often doped with nonmetals such as N, F [65], B [66], and C [67] to improve its photocatalytic antimicrobial properties by modifying the band structure of TiO_2_ [45,68].

Mukherjee et al. [65] prepared N, F-doped TiO_2_ particles with sizes of 175–225 nm. The absorption wavelength is extended to the visible light region (Figure 6A(a,b)) These particles can stably produce ·OH (Figure 6A(c,d)) and inhibit spore germination in a dose-dependent manner, especially under visible light. (Figure 6B). Figure 6C shows images of fungal colonies after 5 days/18 days of treatment with different doses of nanoparticles in the presence/absence of visible light. The nanoparticles had obvious inhibitory effects on fungi at 5 days and 18 days under light conditions.

Boron and its derivatives also enable TiO_2_ to exhibit photocatalytic antibacterial activity [66,69,70]. The B-p and O-2p orbitals can be mixed to reduce the band gap, reduce electron-hole recombination, and effectively improve the photocatalytic activity [71]. Wong et al. [66] prepared a series of B-doped TiO_2_ thin films (117–484 nm; TiO_2_(B)), of which a 484 nm film (484-nm TiO_2_(B)) has the best photocatalytic antibacterial properties (Figure 7A(a–h)). Compared with the dark treatment conditions, the 484-nm TiO_2_(B) film under visible light irradiation had similar killing effects on gram-positive bacteria (*S. aureus* and *Streptococcus pyogenes*) and gram-negative bacteria (*Acinetobacter baumannii* and *E. coli*) (Figure 7B). Under visible light irradiation, a certain amount of carbon-doped TiO_2_ inactivated *Bacillus subtilis* within half an hour, and the killing rates of vegetative cells and spores of *B. cereus, Bacillus thuringiensis,* and *Bacillus anthracis* were higher than those under dark conditions (Figure 7C,D). It can also reduce the toxicity of anthrax lethal toxins by inactivating toxin protein components. Surviving cells under a subbacterial dose of photocatalyst are also more likely to be eliminated by phagocytes [67]. In addition, graphene or carbon nanotubes (CNTs) can also be used as a matrix to composite with TiO_2_, which can also improve the catalytic efficiency [72,73]. Figure 7E [69] shows the antibacterial effect of different samples on *E. coli*. The addition of B and CNTs enhances the photocatalytic antibacterial ability of TiO_2_, and the 3B-TiO_2_-CNT sample has the best photocatalytic antibacterial effect (Figure 7F).

#### 2.3.3. Polymer Doping, Oxides and Others

In addition to the two strategies of metal doping and nonmetal doping, the strategies of oxide and polymer composites are also often used to enhance the photocatalytic antimicrobial performance of TiO_2_ [74,75,76,77].

Polymers often play a role in improving the dispersion of photocatalysts. For example, a polyacrylate shell can make the particle surface have zwitterionic charge, which makes it have good dispersion in aqueous solution. Particles in solution can form colloids and effectively contact the fungal cell membrane [65]. A uniform distribution of TiO_2_ in organic conjugated polymers (CPs) is conducive to the formation of large interfacial regions, which can maximize the utilization of photocatalysts at the minimum cost and reduce the subsequent separation process. In addition, CPs can also resist UV radiation; have excellent electron mobility; improve the overall conductivity, corrosion resistance, and environmental stability of materials; and optimize the organic–inorganic interactions [78]. After TiO_2_ was combined with high-performance microporous polymers (CMP/TiO_2_), its killing effect on *E. coli* and *S. aureus* reached 98.14% and nearly 100%, respectively (Figure 8A). CMP can increase the optical absorption of TiO_2_ to produce more ·OH and ·O_2_^-^ to destroy the cell membrane, which induces the content to leak out and leads to cell death (Figure 8B,C) [75]. Chitosan can be combined with TiO_2_ nanoparticles through its amino and hydroxyl groups to form composite materials with degradation properties, and the absorption spectrum also moves in the visible light direction [79]. Polyaniline has a strong absorption capacity for visible light. TiO_2_ nanoparticles can also be fixed on the surface of polyaniline-coated kapok fibers (PANI-KpF), and the killing ability toward *E. coli* under light can be increased by 30% [80].

Compared with other semiconductor metal oxides, TiO_2_ has the advantages of photochemical stability and nontoxicity and is often used as the main photocatalyst compound with other oxides [81,82]. Ag_2_O [83], ZnO [84], and other oxides can form heterostructures with semiconductors such as TiO_2_. Liu et al. [85] found that the TiO_2_/Ag_2_O heterostructure can kill almost all *E. coli* after 60 min of visible light irradiation, showing strong photocatalytic bactericidal activity (Figure 9A(a,b)). FeO_X_ [86], Cu_2_O [87], and other materials can also be combined with TiO_2_ to achieve antibacterial effects by generating free radicals. Yao et al. [88] synthesized SnSO_4_-modified TiO_2_. Figure 9B shows the antibacterial activity of pure TiO_2_ and different contents of SnSO_4_-TiO_2_ against *E. coli* (a–g) and *S. aureus* (A–G). For 5 mol% SnSO_4_/TiO_2_, the activity against *E. coli* was 93.6% and that against *S. aureus* was 85%. SnSO_4_/TiO_2_ has better antibacterial activity under light than in the dark (Figure 9C).

**Figure 9 materials-15-05820-f009:**
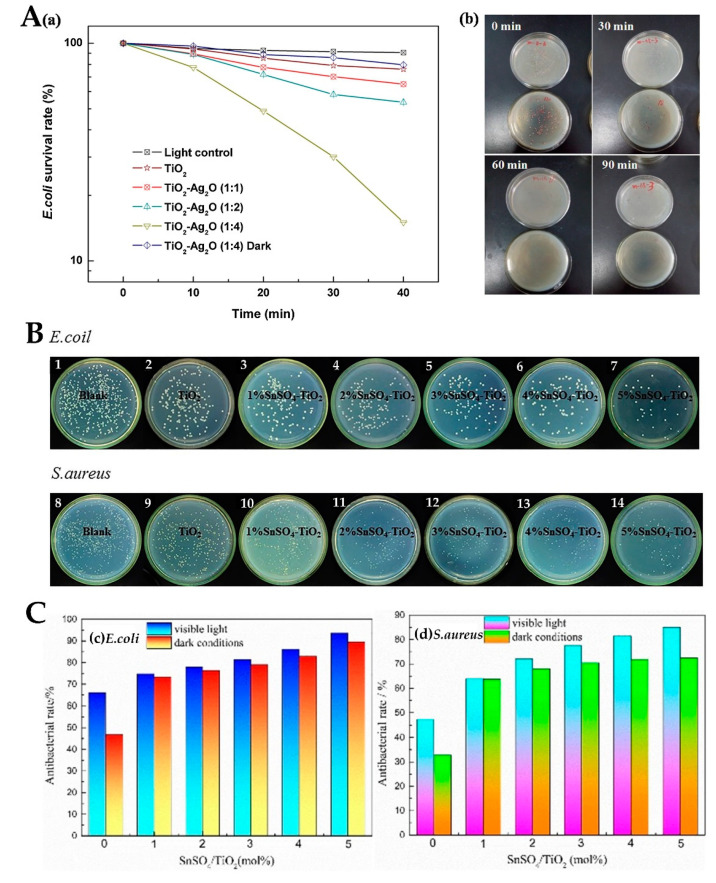
(**A**) Photocatalytic antibacterial efficiency toward *E. coli* with photocatalysts under visible light (**a**) and antibacterial effect of TiO_2_/Ag_2_O (1: 4) on *E. coli* under visible light (**b**) [85]; (**B**) antibacterial activity of different samples against *E. coli* (**1**–**7**) and *S. aureus* (**8**–**14**); (**C**) antibacterial activity of different samples against *E. coli* (**c**) and antibacterial activity of pure TiO_2_ and xSnSO_4_-TiO_2_ against *S. aureus* in the presence/absence of visible light (**d**) [88].

**Table 2 materials-15-05820-t002:** Antimicrobial effect of different TiO_2_ materials under visible light.

Material	Grain Size: nm	Microorganism	Light Source	Antimicrobial Effect	Reference
Au/Pt-TiO_2_	Au (1 min of coating): 9.36 ± 1.88 (diameter), 8.77 ± 1.90 (height) Pt (2 min of coating): 20.72 ± 5.21 (diameter), 30.42 ± 6.01 (height)	*S. aureus*	Visible light (470 nm)	Au TiO_2_ NTs: colony forming unit decreased from approximately 235 to 60 (×10^4^ CFU/mL) Pt TiO_2_ NTs: colony forming unit decreased from approximately 290 to 75 (×10^4^ CFU/mL)	[59]
Ag and Au/TiO_2_	Ag:6 Au:14	*E. coli*	Visible light (>420 nm)	Ag TiO_2_ NTs: almost 100% after 1 h Au TiO_2_ NTs: increased 4 times compared to that under UV irradiation (77.5%)	[89]
Cr-TiO_2_	5.8	*E. coli*	Visible light (360–740 nm)	>95% (100μg/mL)	[62]
Cu, Pt and Ag-TiO_2_	12–15 (catalyst)	*E. coli*	Visible light (>450 nm)	>99%	[58]
Fe-TiO_2_	19.24–22.24	*E. coli*	Visible light (400–700 nm)	97.57% (0.1 at% of Fe-doped TiO_2_ films)	[61]
TiO_2_(B)	117–484 (thickness)	*E. coli, A. baumannii, S. aureus and S. pyogenes*	Visible light (60 W, incandescent lamp)	Approximately 50%	[66]
B-TiO_2_-CNT	12–18	*E. coli*	Visible light (>400 nm)	Almost 100% (3B-TiO_2_-CNT nanocomposites)	[69]
B/Ce-TiO_2_	23	*E. coli and S. aureus*	Visible light	Almost 99% (*S. aureus*) Almost 100% (*E. coli*)	[71]
Ag_2_O-TiO_2_	5–30	*E. coli*	Visible light (>400 nm)	Almost 100%	[83]
Cu_2_O/TiO_2_	8	*E. coli and C. albicans*	Visible light (>420 nm)	>99%	[87]
CuO/TiO_2_	23.6–24.2(anatase), 42.8–44.5(rutile)	*S. aureus*	Visible light	Almost 100%	[90]
Polyaniline-kapok fiber/TiO_2_	6.06	*E. coli*	Visible light	30% less than that in the dark	[91]
Au/ZnO-TiO_2_	10 (Au), 50 (ZnO)	*E. coli*	-	98.2%	[92]
CMP/TiO_2_	650–950	*E. coli and S. aureus*	Visible light (LED)	98.14% (*E. coli*), nearly100% (*S. aureus*)	[75]

## 3. Antimicrobial Mechanism

ROS produced by the material play a major role in the antimicrobial process. The antimicrobial effect of photocatalysts often starts from the broken cell membrane. Then, the antimicrobial effect of photocatalysis can be achieved in many ways; for example, the lipid layer on the cell surface is destroyed, resulting in the leakage and destruction of intracellular substances such as proteins and DNA [93,94,95,96], the destruction of the intracellular electron transport system, and the inactivation of intracellular enzymes [97,98]. Particularly, among many types of TiO_2_-based photocatalysts, TiO_2_-metal composites can release additional metal ions. ROS and metal ions jointly play a role in the antimicrobial process. ROS can attack the structure of cells, while metal ions can contact cells through electrostatic adsorption and other functions, causing denaturation of proteins and other substances [99,100]. ROS are produced by the material itself at the beginning. After the release of metal ions, ROS can also be generated by metal ions. In the following sections, we will discuss the production of ROS and the damage of ROS and metal ions on the cell membrane.

### 3.1. Production and Function of Reactive Oxygen Species

Photocatalysts can produce a variety of ROS, which can cause peroxidation of the cell membrane, produce lipid peroxide, change the fluidity and permeability of the cell membrane, cause cell content to flow out, and cause cell death [101,102,103,104]. ROS may be the most important contributors to antimicrobial activity [105].

#### 3.1.1. Production of ROS

The energy band of a photocatalyst is composed of a valence band (VB) and conduction band (CB), and there is a band gap energy between them. Under the excitation of a certain light energy, the electrons excited in the VB transition to the CB, while leaving holes in the VB, forming electron-hole pairs. Electrons and oxygen molecules generate ·O_2_^−^, and holes and water molecules can generate ·OH, which play a major role in photocatalytic antimicrobial activity. Redox reactions occur on the surface of semiconductors due to the creation of positive holes and negative electrons [106,107,108].

However, electron-hole recombination reduces the efficiency of the carrier, and TiO_2_ can only be activated under UV light, so researchers can improve the photocatalytic efficiency by modifying the materials. For example, SiO_2_/TiO_2_ materials show good adsorption to bacteria and a large specific surface area and can produce · O_2_^−^, ·OH, and H_2_O_2_ to kill cells [50] (Figure 10A). The Fermi level of Ag below the conduction band of TiO_2_ can modify TiO_2_. Doped silver exists on the surface of TiO_2_ and does not affect the crystal structure of TiO_2_ particles. The photogenerated electrons transfer to silver on the surface of TiO_2_, and a Schottky barrier is formed at the Ag/TiO_2_ interface, which hinders the recombination of electron-hole pairs [109,110], promotes interface charge transfer (IFCT) and electron-hole separation efficiency, and improves the capture efficiency of O_2_ [83,111] (Figure 10B).

#### 3.1.2. Function of ROS

Many studies have clearly shown that TiO_2_ nanoparticles can produce various ROS in aqueous solution, such as ·OH and O_2_^-^ and H_2_O_2_ [65,88,112,113]. ·OH and H_2_O_2_ can damage DNA and cellular proteins. The ·OH produced by N and Fdoped TiO_2_ nanoparticles synthesized by Mukherjee et al. [65] can react with the glucan and chitin layers of fungal cell walls, resulting in cell wall degradation and subsequent cell death (Figure 11A). Yao et al. [88] synthesized SnSO_4_-modified TiO_2_, which can first oxidize enzymes and proteins related to the biofilm. After entering the cells, ROS can continue to attack the cells and eventually cause cell death (Figure 11B). The silver ions loaded on the photocatalyst can attack the mitochondrial membrane and cause mitochondrial damage, which can produce more ROS killing cells after damage [114,115].

### 3.2. Release and Function of Metal Ions

The release of metal ions also plays an important role in the antimicrobial process. First, metal ions have high affinity for negatively charged cell membrane surfaces, and they can also produce ROS, which kill the microbial cell membrane and lead to cell membrane dysfunction [98,116]. After the microbial cell membrane is attacked, it breaks, which exposes the organelles inside the cell. Ti^4+^ in TiO_2_ has little contribution to the antimicrobial activity [117], but TiO_2_-loaded Ag [58], Au [89], Cu [58], and Zn [84] can lose electrons to form metal ions, dissolving in the solution. The toxicity of metal ions to cells is related to the morphology of the catalysts and microbial types [49,75,118,119]. For example, Ag has a certain affinity for N and S and can react with proteins and nucleic acids containing thiol and amino groups [120,121,122]. The following figures are schematic diagrams of the effect of metal ions (Figure 12A) and TiO_2_ (Figure 12B) particles on cells, which have certain similarities.

The homeostasis of metal ions is critical to microbial life because they are involved in the regulation of various alternative functions, such as dehydrogenases, cofactors and catalysts, and structural stabilizers of enzymes and DNA-binding proteins [124]. Therefore, some bacteria have developed mechanisms that regulate the inhalation and outflow of metal ions to maintain stable intracellular ion concentrations [125]. However, excess metals or metal ions are toxic to bacterial cells. Metal can be adsorbed on the cell membrane, which destroys the charge balance of the cell membrane and leads to cell deformation [100,126]. For example, in enzyme and protein solutions, ZnO plays a fatal role in the production of ionic signals between various cells and intercellular cytoplasmic instruments [127]. Zeta-positive ZnO nanoparticles can destroy the membrane of gram-negative *E. coli* [128]. Due to their oxidizing ability, zinc ions react not only with organic functional groups (thioglycosides, carboxyl groups, and hydroxyl groups) but also with bacterial cells and membrane proteins. They enter bacterial cells and disrupt their electron transport system, resisting enzyme and protein gene expression functions and thus producing targeted antibacterial effects [97].

## 4. Applications

TiO_2_ and its modified composites show excellent photocatalytic antimicrobial activity after light irradiation. They have broad application prospects in food packaging and preservation [129], self-cleaning fabric [130,131], and so on [129,132].

### 4.1. Food Packaging and Preservation

Microbial contamination is a potential risk factor for food safety and quality, which may lead to outbreaks of foodborne diseases [133]. Heat treatment is a common sterilization method in food, but it may also lead to adverse changes, such as loss of nutrients. Currently, the application of nonthermal treatment in food has attracted increasing attention [134,135,136]. TiO_2_ can be used for food packaging to prevent microbial contamination.

Goudarzi et al. [137] prepared an eco-friendly starch-TiO_2_ composite that can absorb UV light, reducing the damage of singlet oxygen generated by UV radiation to food vitamins and proteins and the oxidation of lipids. TiO_2_ can also be prepared into composite films with k-carrageenan (KC) and konjac glucomannan (KGM). The killing effect of this film on *Penicillium viridicatum* reached nearly 80% after 6 h of illumination, which can be applied to the preservation and storage of strawberry [138] (Figure 13A,B). TiO_2_ can also be used in combination with chitosan to prepare composite films [116]. The film killed more than 90% of four typical foodborne pathogens, *E. coli* (gram-negative bacteria), *S. aureus* (gram-positive bacteria), *C. albicans* (fungi), and *A. niger* (molds), after 4 h of visible light irradiation, especially 99.9% of *E. coli*. After 12 h of treatment, all tested strains were completely killed. This film showed effective resistance to bacteria and fungi, which could be applied to the preservation of grapes, and the preservation period could reach 15 days (Figure 13C). TiO_2_ can be used to control tomato *Fusarium wilt*. As shown in Figure 13D, tomatoes were infected with fungal spores in the marked areas followed by visible-light treatment for 8 days. Tomato soaked in N, F-doped TiO_2_ particles at the wound will not be infected by *Fusarium oxysporum* after 8 days [65].

### 4.2. Self-Cleaning Fabric

In our daily clothing, we inevitably encounter various microorganisms. Cotton fabric has strong moisture absorption and good air permeability, which is conducive to the reproduction of microorganisms [139]. This damages the fabric, and at the same time, the close contact between the fabric and the human body poses a threat to human health [140,141]. Luckily, these bacteria have poor resistance because the environment is suitable for their growth, and self-cleaning clothing can play a better role [142].

Zhu et al. [143] prepared a new antibacterial material ZIF-L@TiO_2_/fabric by mixing TiO_2_ with zeolitic imidazolate framework-L (ZIF-L) and applied it to fabrics. ZIF-L and TiO_2_ can synergistically produce more ROS. As shown in Figure 14A, the LB liquid culture medium treated with ZIF-L@TiO_2_/fabric under irradiation conditions became clearer than that of the other groups. ZIF-L@TiO_2_/fabric can reduce the bacterial viability of *E. coli* to 2% under light, demonstrating a strong antibacterial ability. The ZIF-L@TiO_2_/fabric was placed on LB solid medium and cultured for 5 days, and the antibacterial effect was still obvious (Figure 14B). N, F-doped TiO_2_ can also be applied to cotton fabric surfaces to make light-responsive self-cleaning clothing; N, F-doped TiO_2_ also exhibits excellent surface hydrophobicity, which makes it difficult for water droplets to lubricate its surface [144] (Figure 14C), and Cu-doped TiO_2_ can also effectively remove coffee stains [145] (Figure 14D).

### 4.3. Others

In addition to their application in food packaging and fabric, photocatalytic antimicrobial materials have many applications in other fields. For example, in the field of building materials, TiO_2_ can be used as a semitransparent antibacterial coating. Under UV light irradiation, it showed significant antibacterial activity and resistance to biofilm formation [146,147] (Figure 15A). Figure 15A is a 24 h fluorescence map of the experiment, in which red represents bacteria and image (c-1) was taken at the center of the sample (c-1) and image (c-2) at the edge. It can be seen that the amount of bacteria with the coating is significantly reduced, while more bacteria exist in the part without the coating. In the field of sewage treatment, photocatalytic antimicrobials can also be used as disinfectants for sewage to inhibit microbial production [148,149,150,151]. Under illumination at 25 °C, the sample can exert a certain bactericidal effect on different types in sewage treatment at 25 ℃ [148] (Figure 15B). In the field of medicine, photocatalysts can be used for oral biofilm inhibition [48]. TiO_2_ can also be used in osteogenic materials, which have excellent properties such as antibacterial, osteogenic, and wear resistance [152] (Figure 15C). Under UV irradiation, Ti6Al4V/TiO_2_/SrTiO_3_ had the largest bacteriostatic zone (Figure 15D), and the proliferation rate of *E. coli* cultured on Ti6Al4V/TiO_2_/SrTiO_3_ was only 3%, while as a control, the proliferation rate of Ti6Al4V reached 36%. The antibacterial properties of the material can reduce the risk of bacterial infection during surgery.

## 5. Conclusions

TiO_2_ shows enormous potential in the field of photocatalytic antimicrobials. In this paper, the antimicrobial activities of TiO_2_ photocatalysts under different light sources, especially under visible light, were reviewed based on the source of energy–optical energy. In addition, we also reviewed their main synthesis methods, antimicrobial mechanisms, and application fields. We can reduce the electron-hole recombination, improve the surface properties, and increase the contact area with bacteria by doping metal ions, nonmetal ions and compounding with polymers to extend the photocatalytic activity to the visible light region and enhance the photocatalytic performance. These materials can inhibit microbial activity and achieve antimicrobial effects by producing ROS and hydroxyl free radicals to destroy the integrity of the cell membrane, damage microbial DNA, and inhibit enzyme activity.

However, the photocatalytic antimicrobial mechanism still needs to be further studied. We still need to study the photocatalytic kinetics more systematically, clarify the various effects of modified materials on photocatalytic antimicrobial properties, and connect the photocatalytic properties and antimicrobial properties of the materials more closely so that the materials can be widely used in food packaging, self-cleaning fabrics, sewage treatment, and other fields. The preparation of photocatalysts has a great influence on their photocatalytic antimicrobial performance. The preparation method should be selected depending on the application field, which requires us to have a further understanding of the preparation method and conditions. For nonmetal-doped photocatalysts, although this kind of photocatalyst can modify the band structure of TiO_2_, the nonmetallic elements contained in it can be removed at high temperature, making it difficult to control the content of nonmetallic elements. This puts forward a very high requirement on the preparation conditions. In addition, although TiO_2_-metal composites have an additional ion release effect and polymers can also change the surface properties of particles, their durability, such as polymer aging, should also be considered. It is believed that with the deepening of research, catalysts with high photocatalytic and antimicrobial properties, such as TiO_2_, will be more widely used.

## Figures and Tables

**Figure 1 materials-15-05820-f001:**
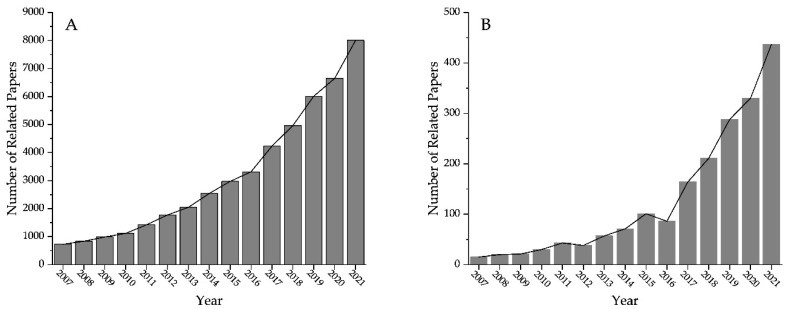
Statistical analysis of the number of related papers published in Web of Science with the keywords: (**A**) ‘Photocatalyst’ and (**B**) ‘Photocatalytic antimicrobial’.

**Figure 3 materials-15-05820-f003:**
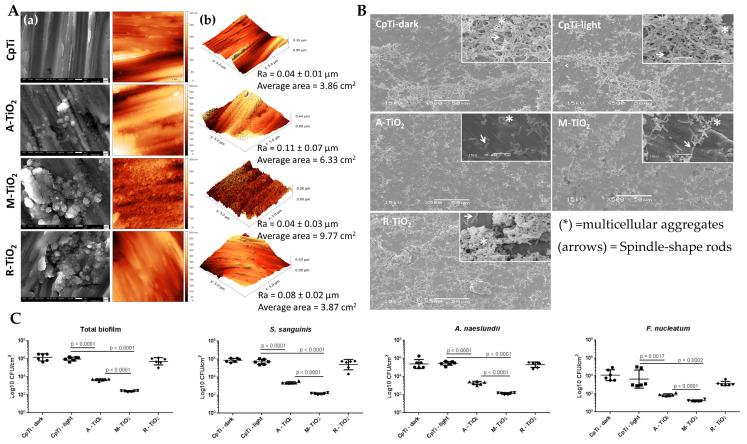
(**A**) SEM (**a**) and AFM (**b**) analyses of different samples; (**B**) SEM micrographs of biofilms formed on different samples; (**C**) Colony-forming units (CFUs) on surfaces of different sample after 1 h of UVA light irradiation [48].

**Figure 4 materials-15-05820-f004:**
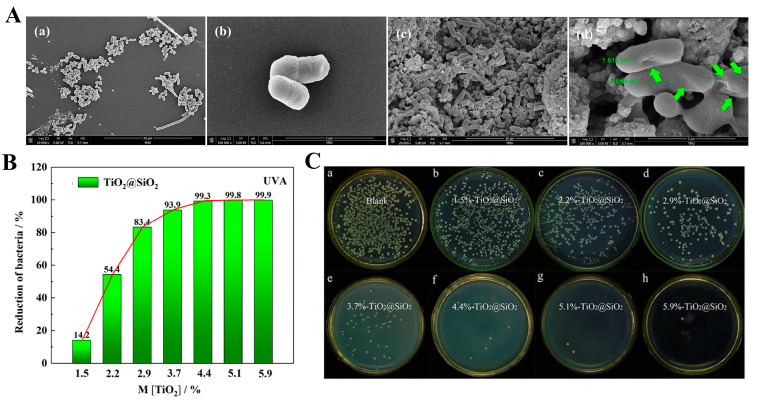
(**A**) SEM images of *E. coli* before (**a**,**b**) and after (**c**,**d**) TiO_2_@SiO_2_ treatment; (**B**) image of the antibacterial effect of TiO_2_@SiO_2_ hybrid materials with different TiO_2_ contents on *E. coli* under UVA irradiation; (**C**) effect of the antimicrobial property of TiO_2_@SiO_2_ hybrid materials under UVA irradiation [50].

**Figure 5 materials-15-05820-f005:**
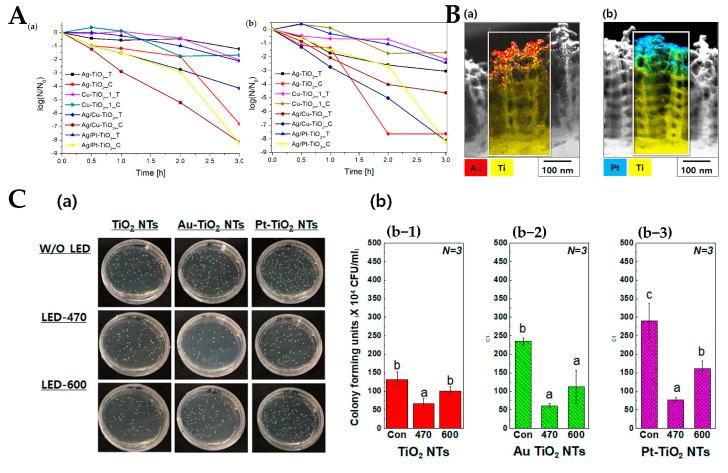
(**A**) antimicrobial activity in the dark (**a**) and under visible light (**b**) (λ > 450 nm) [58]; (**B**) TEM images of Au–TiO_2_ NTs (**a**) and Pt–TiO_2_ NTs (**b**); (**C**) photographs of antibacterial *S. aureus* agar diffusion tests (**a**) and the results of CFUs per unit volume of *S. aureus* cultured on different letters with or without 470 nm and 600 nm visible irradiation (**b**) [59].

**Figure 6 materials-15-05820-f006:**
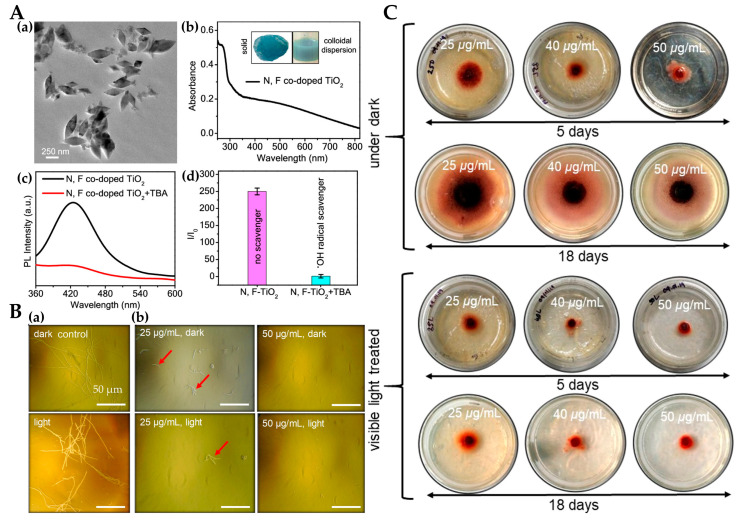
(**A**) TEM image of N, F doped TiO_2_ nanoparticles (**a**); UV−vis absorption spectrum of colloidal nanoparticles (**b**); evidence of reactive oxygen species (ROS) generation by nanoparticles under visible-light exposure (**c**); tert-butyl alcohol (TBA) as a ·OH scavenger to prove that ·OH is the dominant ROS component (**d**); (**B**) microscopic imaging of *Fusarium oxysporum* spore germination under different conditions. Red arrows indicate fungal spores. (**C**) Images of fungal colonies [65].

**Figure 7 materials-15-05820-f007:**
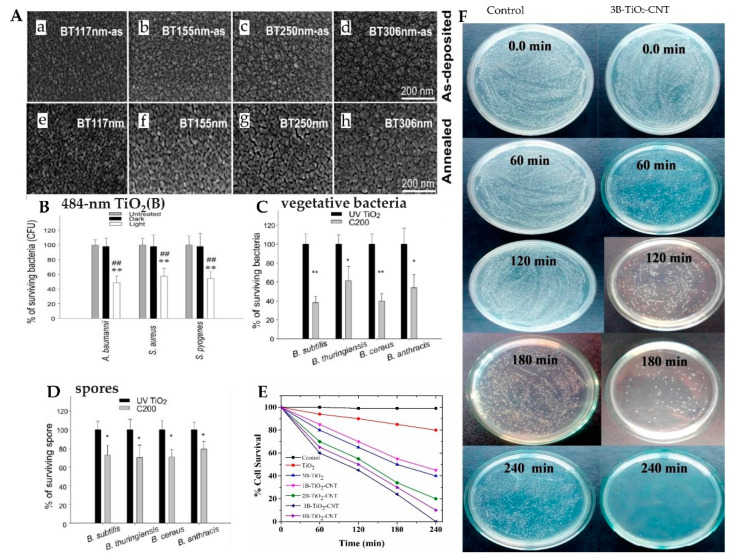
(**A**) SEM image of TiO_2_ (**B**) films with various thicknesses before (**a**–**d**) and after (**e**–**h**) annealing; (**B**) visible light-induced photolytic killing of pathogenic bacteria [66]; (**C**) antibacterial properties of C200 NPs against vegetative bacteria of Bacillus species; (**D**) antibacterial properties of C200 NPs against spores of Bacillus species [67]; (**E**) % survival of *E. coli* with TiO_2_TiO_2_TiO_2_different samples as a function of time under visible light; ## *p* < 0.01, ** *p* < 0.01, * *p* < 0.05. (**F**) photographs of the photoinactivation of *E. coli* under visible light exposure [69].

**Figure 8 materials-15-05820-f008:**
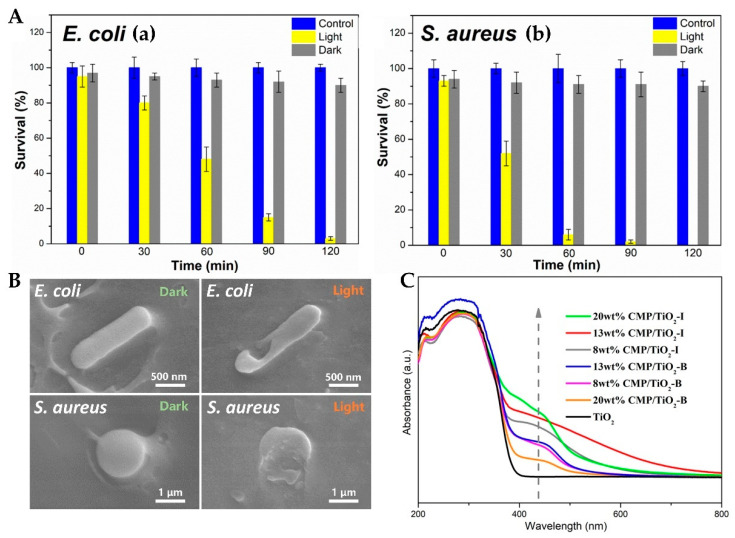
(**A**) Photocatalytic inactivation of *E. coli* (**a**) and *S. aureus* using 13 wt% CMP/TiO_2_-I nanocomposites. Growth images of *E. coli* (**b**) and *S. aureus* on agar plates; (**B**) SEM images of bacteria (incubated with 13 wt% CMP/TiO_2_-I nanocomposites under light irradiation and in the dark); (**C**) DRS of samples [75].

**Figure 10 materials-15-05820-f010:**
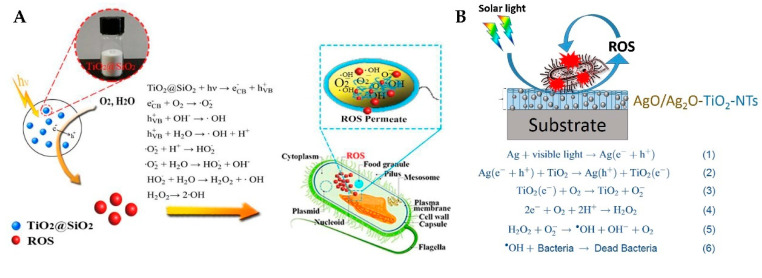
Schematic of photocatalytic antimicrobial mechanism of TiO_2_@SiO_2_ hybrid materials and bacterial inactivation on Ag decorated TiO2-NTs [50,111].

**Figure 11 materials-15-05820-f011:**
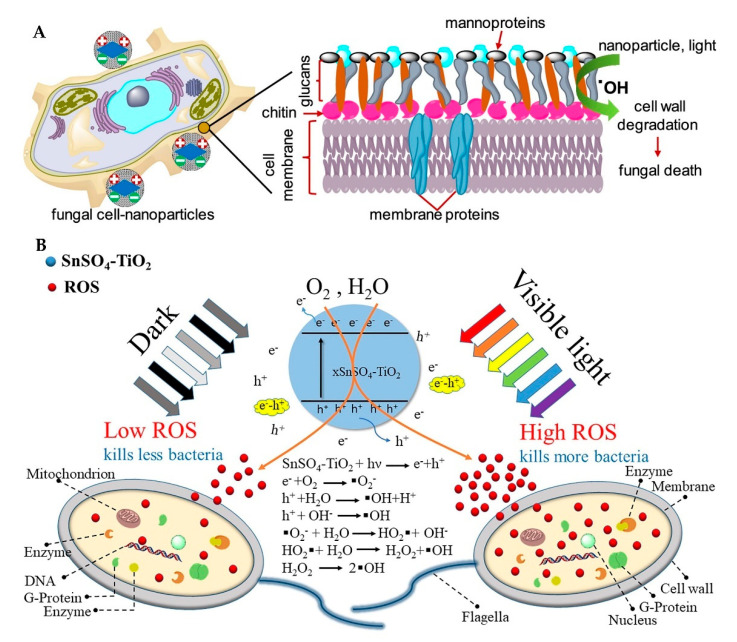
(**A**) Proposed mechanism of the antifungal activity of N, F doped TiO_2_ nanoparticles [65]; (**B**) schematic diagram displaying the antibacterial mechanism of SnSO_4_-TiO_2_ [88].

**Figure 12 materials-15-05820-f012:**
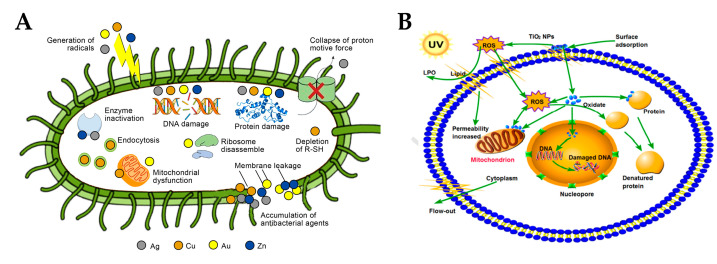
Schematic diagrams of the effect of metal ions (**A** [121]) and TiO_2_ (**B** [123]) particles on cells.

**Figure 13 materials-15-05820-f013:**
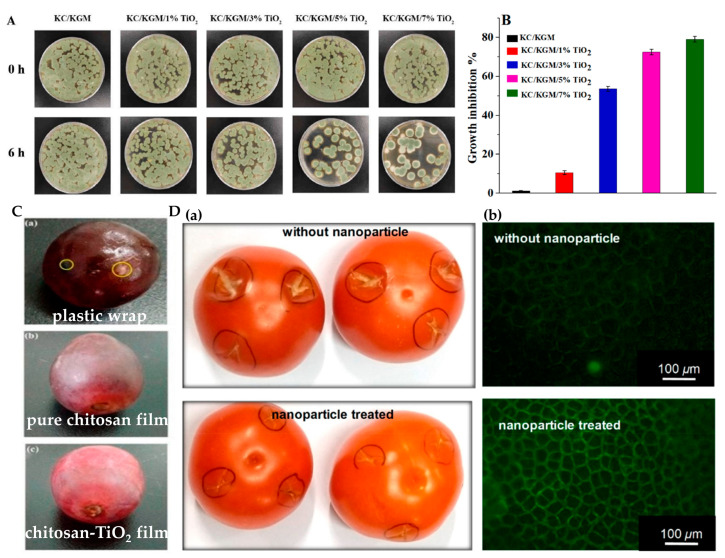
(**A**) Effect of the prepared composite films on the growth of *P. viridicatum* after 0 h and 6 h of irradiation; (**B**) effects of TiO_2_ content on the inhibition efficiency of KC/KGM/TiO_2_ nanocomposite films against *P. viridicatum* [138]; (**C**) preservation of red grapes packed in different materials at 37 °C for 6 days [116]; (**D**) (**a**) in vivo fungitoxicity and infection control by nanoparticles. (**b**) Evidence of intact immunity of nanoparticle and visible-light-treated tomato fruit via nitric oxide imaging. Green fluorescence indicates the presence of nitric oxide [65].

**Figure 14 materials-15-05820-f014:**
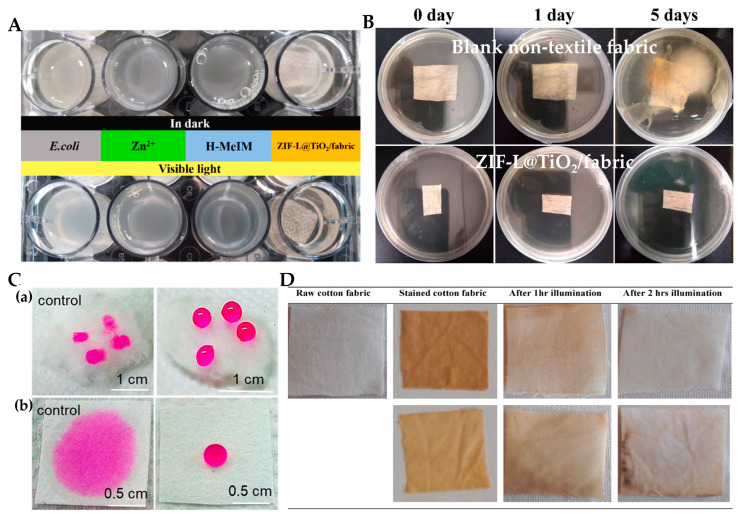
(**A**) Photographs of the antibacterial activity of Zn2+, H-MeIM and ZIF-L@TiO_2_/fabrics against *E. coli*; (**B**) photographs of bacterial growth on blank nontextile fabric and ZIF-L@TiO_2_/fabric under natural light for 1–5 days at 37 °C [143]; (**C**) nonwetting behavior of modified cotton (**a**) and paper (**b**) surface [144] paper; (**D**) self-cleaning performance of 1% Cu (II)-doped TiO_2_ calcined at 25 °C [145].

**Figure 15 materials-15-05820-f015:**
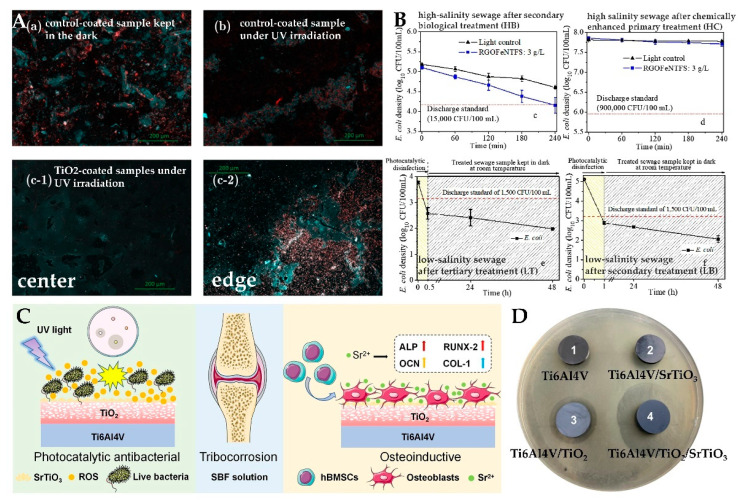
(**A**) Epifluorescence images of surface samples after 24 h of the experiment (blue color represent the coating and red to highlight the bacteria) [146]; (**B**) photocatalytic disinfection performance of sewage samples and the light control tests of (**c**) HB and (**d**) HC under simulated solar light at 25 °C; and regrowth test of inactivated bacteria (24 h and 48 h after reaction) in (**e**) LT and (**f**) LB [148]; (**C**) schematic diagram of TiO_2_/SrTiO_3_ function; (**D**) bacteriostatic circles of different samples after 24 h [152].

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
