# Peer review of "Preparation, Antimicrobial Properties under Different Light Sources, Mechanisms and Applications of TiO2: A Review"

_materials, 2022, doi:10.3390/ma15175820_

Round 1
Reviewer 1 Report
Reviewer Report
This manuscript is reporting on a review of photochemical application of titanium dioxide and its analogues for antimicrobial usages. Shanga et al. summarized the background, preparation, and application of photocatalyst TiO2. This manuscript well describes the related works. As such, after addressing several issues, the manuscript will be published in Materials.
Major Issues
1) In the introduction part, the authors mentioned TiO2 as a promising candidate for photochemical therapeutic materials. This metal oxide is indeed historically important material for this kind of application, and they all are fundamentally based on several breakthrough findings of photochemical oxidation by TiO2 reported in 1938 and “photoelectrochemical” water splitting activity of TiO2 reported in early 1970s. Mentioning them is also very useful for readers to understand why TiO2 is so active and useful, especially under aqueous condition. Distinguishing TiO2 from other metal oxides will also be instructive.
2) in the conclusion part, there is a description “…are reviewed innovatively.” Then, what’s the take-home message?
Minor Issues
1) The authors mentioned “doping with different ions” for improving the photocatalytic performance of titanium dioxide. However, after carefully checked, some of the systems are not “doped,” but just being with co-catalysts (like Ag on TiO2).
2) Similarly, there is a sentence “Liu et al. [72] found that Ag2O-doped TiO2 can kill almost all E. coli” in line 263, but this report was for a bi-photocatalytic system rather than “doped” system. Please check the XRD in Figure 1d of reference [72]. The authors in reference [72] indeed referred to their system as “TiO2/Ag2O composite” or “heterostructure” rather than “doped.” Of course, as was discussed in section 4.2 about the potential role of ion-dissolution, the role of additional metals is not decisive, but at least they are not “doped.” Please clearly separate “doped” and “hetero-structured” systems.
3) in page 1, sentence until line 33 and sentence from 34 are logically less linked.
4) several figures with poor resolution. For example, it’s almost impossible for readers to understand the letters and words in some figures.
5) in line 169, this review mentioned a work for “Cr-doped TiO2” with highly improved performance. What’s the role of Cr?
6) in line 205, what’s TBA?
7) Cite more latest works. Now 2022.
8) Figure 5A, too small to understand the contents.
9) In section 2.3.2, there is a description “their surface chemical properties can effectively interact with fungi.” As is disused in section 4.2, the use of transition metals would have a concern of the potential role of ion-dissolution, while such a non-metal doping system is more appealing (abundant and cheap, and free from such a concern), and the understanding of such an interaction and special effects will be of great importance for future works. As such, more detailed explanation will also be appealing to the readers, if any.
Reviewer 2 Report
The present work makes a detailed review of the antimicrobial properties of TiO2 photocatalysts under different light sources. The work presents some points that urgently need to be improved before publication. Below are my suggestions for improvement:
i) The introduction should be improved; a paragraph was missing outlining how the topics of the present work will be presented;
ii) Images need to be better resolution and/or enlarged. Some figures are illegible, and others are tiny;
iii) In subsection 2.1 Synthesis Methods it is necessary to add the following section 2.1.5 Atomic Layer Deposition (ALD) Method (https://doi.org/10.1016/j.apsusc.2017.05.254; https://doi.org /10.1016/j.apsusc.2021.151891; https://doi.org/10.3390/mi12060588). After this addition, the work will be better.
iv) In sections 2.2 and 2.3, I also see the need to add TiO2 deposited by ALD to make the work more complete;
v) Conclusion should be improved as well.
Reviewer 3 Report
The article Antimicrobial Properties of TiO2 Photocatalysts under Different Light Sources, is a bibliographic review of the use of TiO2 and its derivatives as antibacterial.
The topic of this review is currently of great interest, as the authors indicate in the manuscript, so their approach is considered very timely.
In my opinion the number of figures from other works is very high and they are not corrrctly referenciated. This could be a great problem.
I consider that many aspects of the manuscript should be improved and I suggest a major revision in the aspects detailed below.
1. The word review must appear in the title
2. The focus of the review is not what is expected when reading the title, it should be adapted. There is no focus in relation to the different light sources.
3. In general, the manuscript collects many works but the authors make few contributions of their own. The authors must identify the relevant known, unknown and controversial aspects of the reviewed topic.
4. The organization of the manuscript should be improved by adding epigraphs.
5. Aspects such as the manufacture of composite materials or the application of TiO2 in materials are not covered.
6. The conclusions should be expanded, with a greater analysis of the challenges in this field.
7.-There are many images and figures from other works. The authors must include th refenrences correctly. The figure captions are very complex and the reference does not appear.
Round 2
Reviewer 2 Report
The authors improved the article according to the suggestions. I believe that in this form, the article can be published.
Reviewer 3 Report
The authors have improved the manuscript. I suggest to accept in present form.